# *Diplostomum cf. vanelli* Yamaguti, 1935 (Trematoda: Diplostomidae Poirier, 1886): Morpho-Molecular Data and Life Cycle

Anna V. Izrailskaia [1,*], Vladimir V. Besprozvannykh [1] and Michael Yu. Shchelkanov [2]

1 Federal Scientific Center of the East Asia Terrestrial Biodiversity, Far Eastern Branch, Russian Academy of Sciences, 100-Letiya Street, 159, 690022 Vladivostok, Russia; besproz@biosoil.ru
2 G.P. Somov Research Institute of Epidemiology and Microbiology, Russian Federal Service for Surveillance on Consumer Rights Protection and Human Wellbeing, Selskaya Street. 1, 690022 Vladivostok, Russia; adorob@mail.ru
* Correspondence: anna.kharitonova92@yandex.ru

**Abstract:** Furcocercariae, of the trematodes from the family Diplostomidae, were found in freshwater snails—*Radix auricularia*, which were collected in a reservoir located on Popov Island (Peter the Great Bay, Sea of Japan). The life cycle was experimentally reproduced for the first time, while morphometric data for the development stages were studied and described for the newly discovered trematode. Moreover, molecular data for nuclear and mitochondrial markers were also obtained. It was determined that the morphometric characteristics of the trematode coincided with the species *Diplostomum cf. vanelli*, the molecular data analysis validates the species independence. Furthermore, the study highlights the issue of species identification in the *Diplostomum* genus.

**Keywords:** Trematoda; Diplostomidae; *Diplostomum cf. vanelli*; life cycle; morphological description; 28S; COX1





## 1. Introduction

Trematodes of the genus *Diplostomum* Nordmann, 1832 are a cosmopolitan group of parasites. According to the World Register of Marine Species (WoRMS) (https://www.marinespecies.org/index.php (accessed on 10 April 2024)), this group includes 27 species of fish ocular parasites. A significant number of publications have been devoted to the study of representatives of this genus, concerning both the morphology of developmental stages and issues of taxonomy and systematics [1–10]. Interest in this group of trematodes is determined not only by the solution of taxonomic problems, but also by its high ecological significance. A major feature of the life cycles of species of *Diplostomum* is the metacercarial stage infecting the eyes of fish, often causing blindness and sometimes death. Fry are most susceptible to infection. The extensiveness and intensity of the invasion quickly increases, which leads to the death of the fish if it is not eaten by the final host. Infection can lead to large-scale epizootics among fish, both on natural conditions and in fish farms. In the south of the Russian Far East, six species of *Diplostomum* are currently recorded: *D. kronschnepi* Bychowskaja-Pawlowskaja, 1953, *D. mergi* Dubois, 1932, *D. commutatum* (Diesing, 1850) Dubois, 1937, *D. spathaceum* (Rudolphi, 1819) Olsson, 1876, *D. helveticum* (Dubois, 1929), and *D. vanelli* Yamaguti, 1935 [11–13]. For two of the listed species (*D. spathaceum* and *D. mergi*), molecular data were obtained for samples collected in Ukraine and China, respectively.

During our examination of freshwater malacofauna in a reservoir on the territory of Popov Island, the pulmonated snails *Radix auricularia* (Linnaeus, 1758) (Lymnaeidae Rafinesque, 1815) were found, releasing furcocercariae morphologically similar to Diplostomidae Poirier, 1886. The life cycle was experimentally studied, and morphometric data on development stages and molecular characteristics were obtained to clarify the taxonomic affiliation of the discovered trematodes.

## 2. Materials and Methods

### 2.1. Life Cycle and Morphology of Worms

The snails *R. auricularia*, releasing morphologically identical furcocercariae, were collected manually and using a hydrobiological sieve in a reservoir located on Popov Island in the Peter the Great Bay of the Sea of Japan. Among the 100 collected snails, 10 were infected by Diplostomidae. To determine a second intermediate host, water containing cercariae obtained from one individual of *R. auricularia* was poured into two containers with a volume of 1 L each, of which the first contained 10 tadpoles of Dybowski's frog *Rana dybowskii* Günther, 1876, and the second seven fish, Lake minnow *Phoxinus percnurus* (Pallas, 1814). Tadpoles and fish were caught in a reservoir that did not contain a source of trematode infestation. Before starting experimental infection, 20 tadpoles and 10 fish were dissected from this reservoir for control, with no trematode metacercariae found. Infection of animals was repeated twice, after the fish and tadpoles were kept in separate aquariums with volumes of 10 and 1.5 L, respectively. On the third- and forty-third-day post exposure (PE), metacercariae morphologically similar to those of *Diplostomum* were found in the vitreous body of the eyes of the infected fish using a light microscope. The fish ($n = 5$) were cut into pieces and fed to a two-week-old duckling *Anas platyrhynchos* Linnaeus, 1758 ($n = 1$) along with a portion of food. Nine days PE, the bird was euthanized and necropsied. Six worms were removed from the duckling's intestines.

All procedures contributing to this work comply with the ethical standards of the relevant national and institutional guides on the care and use of laboratory animals including fish, tadpoles, and birds. Euthanasia of all the animals was carried out in accordance with the decision of the Committee on the Ethics of Animal Experiments, Federal Scientific Center of the East Asia Terrestrial Biodiversity (FSCEATB), Far Eastern Branch, Russian Academy of Sciences (FEB RAS) (Permit No. 1 of 25 April 2022) and was done in accordance with ethical standards of the relevant national and institutional guides.

The morphology of live cercariae and metacercariae was examined using light microscopy. Cercariae were fixed in a hot 4% formalin solution before measurements; the metacercariae were measured live. Adult trematodes were fixed in 70% ethanol and then transferred to 96% ethanol for storage. Whole mounts of adult specimens were prepared by staining with alum carmine, dehydration in a graded ethanol series, clearing in clove oil, and mounting in Canada balsam. Measurements of cercariae, metacercariae, and adult stages were made using an ocular micrometer and light microscopy. All measurements were in micrometers (μm). Morphological drawings were made using a drawing tube.

### 2.2. Molecular Data

#### 2.2.1. DNA Extraction, Amplification, and Sequencing

Genomic DNA was extracted from one adult worm using HotSHOT [14]. Partial sequences of the 28S rRNA gene (*28S*) of nuclear DNA and partial sequences of the COX1 gene of mitochondrial DNA were amplified by polymerase chain reaction (PCR) using specific primers (Table 1). Annealing temperatures were 55 and 47 °C for *28S* and COX1, respectively. The efficiency and contamination of PCR were tested by setting positive and negative controls, respectively. The PCR products were sequenced via the Sanger method using a BigDye Terminator Cycle Sequencing Kit (Applied Biosystems, Waltham, MA, USA). Nucleotide sequences were determined on an ABI 3500 Genetic Analyzer (Applied Biosystems, USA) at the FSCEATB FEB RAS, Russia. Both external and internal primers were used for sequencing (Table 1).



**Table 1.** PCR primers used in amplification and sequencing partial *28S* and COX1 gene sequences from adult *Diplostomum* specimen collected from duckling.

| DNA Region | Primer | Sequence 5′→3′ | Direction | Reference |
|---|---|---|---|---|
| *28S* | digl2 | AAGCATATCACTAAGCGG | forward, external | [15] |
| | 1500R | GCTATCCTGAGGGAAACTTCG | reverse, external | [15] |
| | 900F | CCGTCTTGAAACACGGACCAAG | forward, internal | [15] |
| | 1200R | CTTGGTCCGTGTTTCAAGACGGG | reverse, internal | [15] |
| COX1 | MplatCOX1dF | TGTAAAACGACGGCCAGTTTWCITTRGATCATAAG | forward, external | [16] |
| | MplatCOX1dR | CAGGAAACAGCTATGACTGAAAYAAYAIIGGATCICCACC | reverse, external | [16] |

### 2.2.2. Analysis of Genetic Data

Processing and alignment of consensus sequences were carried out using the FinchTV 1.4 and MEGA 5.0 programs [17]. Phylogenetic relationships within the family Diplostomidae were assessed independently for both markers using newly obtained sequences and data for all species of *Diplostomum* available in GenBank. Sequences less than 1000 bp in length were removed from the analysis. The *p*-distances between and within the species were analyzed using the MEGA software without including indels. Members of the trematode family Diplostomidae, *Postharmostomum commutatum* (Diesing, 1858) (Brachylaimoidea (Joyeux & Foley, 1930)), located left (basal) in the phylogenetic tree based on lsrDNA- and ssrDNA-combined sequences inferred by [18], were selected as outgroups in our study to take root of the reconstructions. The list of samples used in the study is provided in Tables 2 and 3.

Phylogenetic relationships were reconstructed using the Bayesian algorithm in the MrBayes 3.1.2. program [19] by applying a model selected as optimal on the basis of the Akaike information criterion (AIC) in the jModeltest 2.1.7 program [20]: TPM3uf+I+G for *28S*, and TIM3+I+G for COX1. The method of a posteriori probabilities was used for the trees constructed by the Bayesian algorithm. In the Bayesian analysis, 600,000 and 31,300,000 generations of the Markov chain Monte Carlo posterior part third (MCMC) were simulated for *28S* and COX1, respectively. The number of generations was determined to be sufficient since the SD value calculated was <0.01. The selection was performed with a frequency of every 100 and 1000 generations for *28S* and COX1 markers, respectively. Of the samples assessed, 25% were excluded to permit the construction of consensus trees.

**Table 2.** Sequences for the *28S* gene for species of *Diplostomum* used for the phylogenetic analysis of *D. cf. vanelli*.

| Species | Sources | Genbank Accession Number |
|---|---|---|
| *Diplostomum cf. vanelli* | this study | PP002326 |
| *Diplostomum alascense* | [21] | MZ314153 |
| *Diplostomum phoxini* | [18] | AY222173 |
| *Diplostomum alarioides* | [21] | MZ314152 |
| *Diplostomum scudderi* | [21] | MZ314170 |
| *Diplostomum marshalli* | [21] | MZ314167 |
| *Diplostomum gavium* | [21] | MZ314154-55 MZ314157 |
| *Diplostomum pseudospathaceum* | [22] | KR269766 |
| *Diplostomum rauschi* | [21] | MZ314169 |
| *Diplostomum indistinctum* | [21] | MZ314161-65 |
| *Diplostomum spathaceum* | [22] [21] | KR269765 MZ314171 |

**Table 2.** *Cont.*

| Species | Sources | Genbank Accession Number |
|---|---|---|
| *Diplostomum huronense* | [21] | MZ314160 |
| *Diplostomum baeri* | [23] | OK631869 |
| *Diplostomum ardeae* | [24] | MT259036 |
| *Postharmostomum commutatum* * | [25] | MH915390 |

* Outgroup.

**Table 3.** Sequences for the COX1 gene for species of *Diplostomum* used for the phylogenetic analysis of *D. cf. vanelli*.

| Species | Sources | Genbank Accession Number |
|---|---|---|
| *Diplostomum cf. vanelli* | this study | PP002346 |
| | [21] | MZ323257-61 |
| *Diplostomum huronense* | [26] | GQ292488-90 |
| | [8] | KR271069; KR271071-74 |
| | [27] | HM064667-68; HM064672 |
| *Diplostomum rauschi* | [21] | MZ323270 |
| | [21] | MZ323262-63; MZ323265 |
| *Diplostomum indistinctum* | [26] | GQ292482 |
| | [27] | HM064673 |
| | [28] | KT831379 |
| | Unpublished | MF142161; MF142171-73; MF142176; MF142178; MF142184; MF142186-89; MF142191; MF142196; MF142198; MF142209; MF142211-12; MF142214; MF142216; MF142222-24 |
| *Diplostomum baeri* | [29] | |
| | [8] | MH368850-53 |
| | [30] | KR271040; KR271042-47; KR271050-60; KR271062; KR271064-67 |
| | [23] | KM212030-KM212031 |
| | | OK632471-74; OK632476-77 |
| | [21] | MZ323274; MZ323276; MZ323278-81 |
| *Diplostomum spathaceum* | [8] | KR271414-16; KR271419-24; KR271427-28; KR271431-45; KR271449; KR271451; KR271454-62; KR271465; KR271467-69 |
| | [22] | KR269763 |
| *Diplostomum alascense* | [21] | MZ323250 |
| *Diplostomum scudderi* | [21] | MZ323273 |
| *Diplostomum alarioides* | [21] | MZ323249 |
| *Diplostomum pseudospathaceum* | [22] | KR269764 |
| | [8] | KR271083-89; KR271091-92 |
| *Diplostomum gavium* | [21] | MZ323251; MZ323255 |
| *Diplostomum marshalli* | [21] | MZ323268 |
| *Diplostomum lunaschiae* | [24] | MT324602; MT324607-08; MT324612-14; MT324617; MT324621-22; MT324595-96; MT324599 |
| *Diplostomum ardeae* | [24] | MT324592-93 |
| | [8] | KR271033 |
| *Diplostomum mergi* | [8] | KR271082 |
| | Unpublished | KY271543 |
| *Postharmostomum commutatum* * | [31] | MN200359 |

* Outgroup.

## 3. Results

*3.1. Morphology Data*

*Diplostomum cf. vanelli*

A total of six worms, with a well-developed reproductive system but without eggs in the uterus, were found from the duckling's intestines.

Definitive host: *Anas platyrhynchos* (experimentally).

Site: small intestine.

First intermediate host: *Radix auricularia*.

Second intermediate host: *Phoxinus percnurus* (experimentally).

Site: Eye vitreous humor.

Locality: Lake on Popov Island (Vladivostok city), Primorsky Region, the Russian southern Far East, 42°57′ N, 131°43′ E.

Type-deposition: Paratype No. 251-255-Tr. This material is held in the parasitological collection of the Zoological Museum (Federal Scientific Center of the East Asia Terrestrial Biodiversity, Far East Branch of the Russian Academy of Sciences, Vladivostok, Russia); e-mail: petrova@biosoil.ru. Deposited: 22 September 2023.

*Adult worm* (five experimentally-derived specimens) (Figure 1A,B, Table 4). Body, distinctly bipartite. Prosoma spatulates, without ventral concavity. Opisthosoma elongates. Oral sucker subterminal, round or transversely oval, equal to or slightly less than ventral sucker. Pseudosuckers two, well-developed, lateral, on either side of oral sucker. Pharynx well-developed, round, adjacent to oral sucker. Intestinal bifurcation in middle third of forebody. Oesophagus short. Caeca short, terminates before reaching posterior end of body. Ventral sucker round, located just anterior to holdfast organ or partially covered by its front edge. Holdfast organ large, circular, with median longitudinal slit. Testes tandem, located in middle part of opisthosoma. Anterior testis asymmetrical. Posterior testis symmetrical, dumbbell-shaped. Seminal vesicle curved, posterior to testes. Ovary round or oval, situated in front of the anterior testis, dextral to midline. Vitelline fields contain small irregular follicles. In prosoma, vitelline follicles surround holdfast organ, then aggregate into six longitudinal ribbon fields (three on each side of median line), which extend to level of posterior end of pharynx. In opisthosoma, vitelline follicles occupy ventral side of body, and terminate before reaching end of opisthosoma. Copulatory bursa opened dorsally, located near posterior end of body. Eggs undetected in uterus.

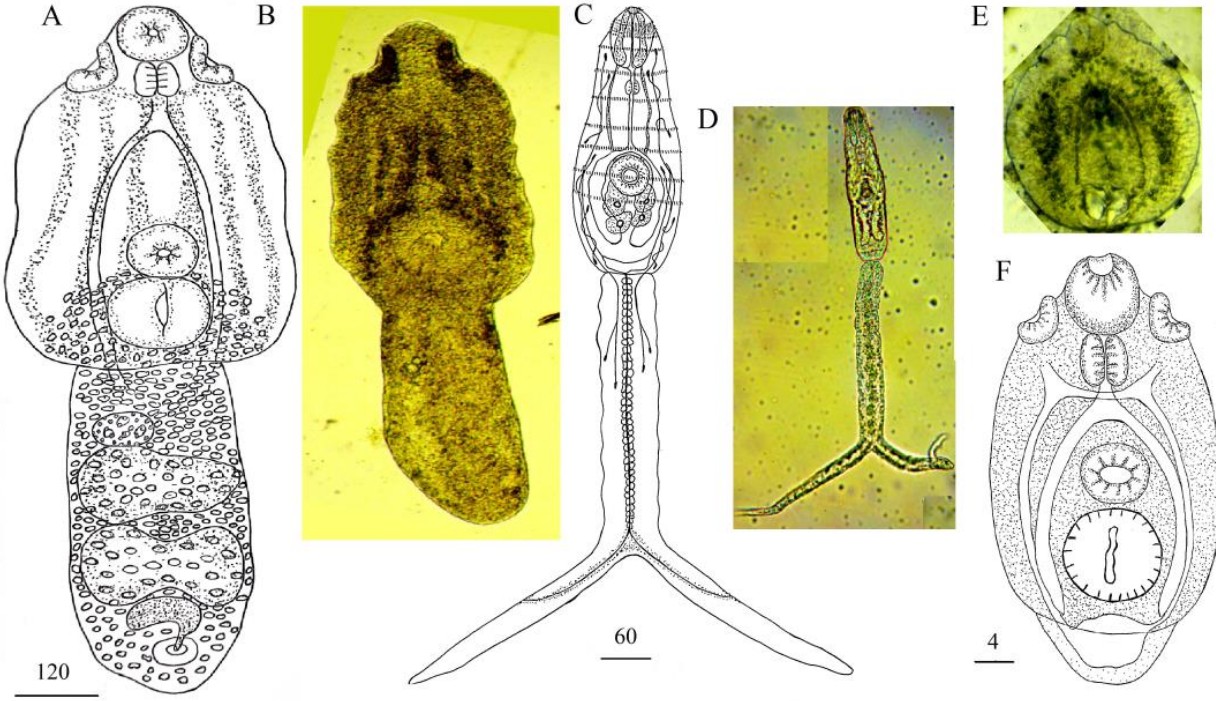

**Figure 1.** *Diplostomum cf. vanelli*: (**A,B**) adult worm; (**C,D**) cercaria; (**E,F**) metacercaria. Scale bars: μm.

*Cercaria* (based on 15 specimens) (Figure 1C,D). Body, 212–300 × 62–87, elongated oval, filled with numerous granular formations. Anterior end of body armed by three to four rows of transverse spines surrounding oral organ. At level of middle part of oral organ, five to six rows of spines, close in proximity to each other. Posterior to these rows of spines, seven rows of transverse spines reaching level of posterior end of ventral sucker or slightly posterior to ventral sucker. Oral organ elongated oval, 55–67 × 25–47. Prepharynx short, 4–5. Pharynx small oval, 12 × 10. Caeca reach excretory bladder. Ventral sucker round,

30–42 × 30–45, with spines arranged in two rows. Glands present in two pairs of cells located posterior to ventral sucker. Ducts of glands open at anterior end of body. Excretory system includes small excretory bladder, first-order ducts, second-order ducts, and caudal duct. Caudal duct, splits anterior to furcae into two canals reaching middle part of furcae, where it opens with pores. Flame cell formula 2[(1 + 1 + 1) + (1 + 1 + 1 + [2])] = 16. Tail stem, 225–282 × 42–50. Length of furcae equal to length of tail stem, 207–287 × 12–22.

*Metacercaria* (based on 15 specimens) (Figure 1E,F). Body consists of large prosoma (390–450 × 210–280) and small opisthosoma. Prosoma is leaf-shaped, gray due to pigmentation. Oral sucker, 70–90 × 50–90, round, subterminal. Pseudosuckers two, well-developed, lateral, on either side of oral sucker, 40–60 × 30–40. Prepharynx absent. Pharynx, 50–60 × 30–50, round or oval. Oesophagus long. Intestinal bifurcation on the border of anterior and middle thirds of body. Caeca reach anterior edge of excretory bladder. Ventral sucker, 30–60 × 30–90, round, or transversely oval, anterior to holdfast organ. Holdfast organ oval, 100–120 × 110–120. Four ducts diverge from excretory bladder, two of them blind and reach holdfast organ, and the other two form longitudinal channels connected transverse commissures forming network of channels. Number of excretory bodies from 150 to 180 calcareous bodies.

**Table 4.** Measurements of adult *Diplostomum vanelli* of study by Yamaguti, 1935 cit. [32], Dubois, 1938 cit. [32], and *Diplostomum cf. vanelli* of present study (μm).

| | *Diplostomum vanelli* | | |
|---|---|---|---|
| **Source** | **Present Study** | **Yamaguti 1935 cit. [32]** | **Dubois 1938 cit. [32]** |
| Body length | 785–1145 | 1210–1600 | 1200–1500 |
| Prosoma length | 410–595 | 680–800 | 670–750 |
| Prosoma width | 250–370 | 450–650 | 480–520 |
| Opisthosoma length | 375–550 | 530–800 | 580–750 |
| Opisthosoma width | 175–245 | 320–470 | 400–470 |
| Oral sucker length | 50–80 | 48–75 | 55–62 |
| Oral sucker width | 45–100 | 38–84 | 62–79 |
| Lateral pseudosucker length | 55 | 90–102 diameter | 72–100 diameter |
| Lateral pseudosucker width | 30–80 | | |
| Pharynx length | 50–70 diameter | 54–66 | 53–65 |
| Pharynx width | | 42–54 | 45–50 |
| Ventral sucker length | 55–85 | 72–108 diameter | 74–82 |
| Ventral sucker width | 70–105 | | 86–90 |
| Holdfast organ length | 80–95 | 150–208 diameter | 220–280 diameter |
| Holdfast organ width | 100–125 | | |
| Anterior testis length | 100–220 | 110–180 | 140–160 |
| Anterior testis width | 70–130 | 200–330 | 270–305 |
| Posterior testis length | 175–220 | 140–200 | 150–205 |
| Posterior testis width | 60–115 | 280–360 | 315–360 |
| Ovary length | 50–60 | 50–110 | 105 |
| Ovary width | 55–75 | 75–160 | 125–140 |
| Eggs length | - | 93–108 | 96–104 |
| Eggs width | - | 54–60 | 54–63 |
| Prosoma/opisthosoma length ratio | 1.0 | - | 0.83–1.0 |
| Oral/ventral sucker length ratio | 0.9 | - | - |

### 3.2. Molecular Data

For trematodes in this study, after alignment, the lengths of the obtained nucleotide sequences were as follows: 1115 bp for *28S* and 405 bp for COX1.

In the phylogenetic reconstruction based on the nuclear marker (Figure 2), the *D. cf. vanelli* specimens obtained in the present study clustered with the specimens designated as *D. spathaceum*, *Diplostomum huronense* (La Rue, 1927) Hughes, 1929, *Diplostomum baeri* Dubois, 1937, and *Diplostomum ardeae* Dubois, 1969. In the phylogenetic reconstruction

based on the mitochondrial marker (Figure 3), the *D. cf. vanelli* specimens obtained in the present study clustered with the specimens designated as *Diplostomum indistinctum* (Guberlet, 1923) (Branch 4). Genetic distances between the discovered trematode and species from the same clade and within clade are presented in Tables 5 and 6.

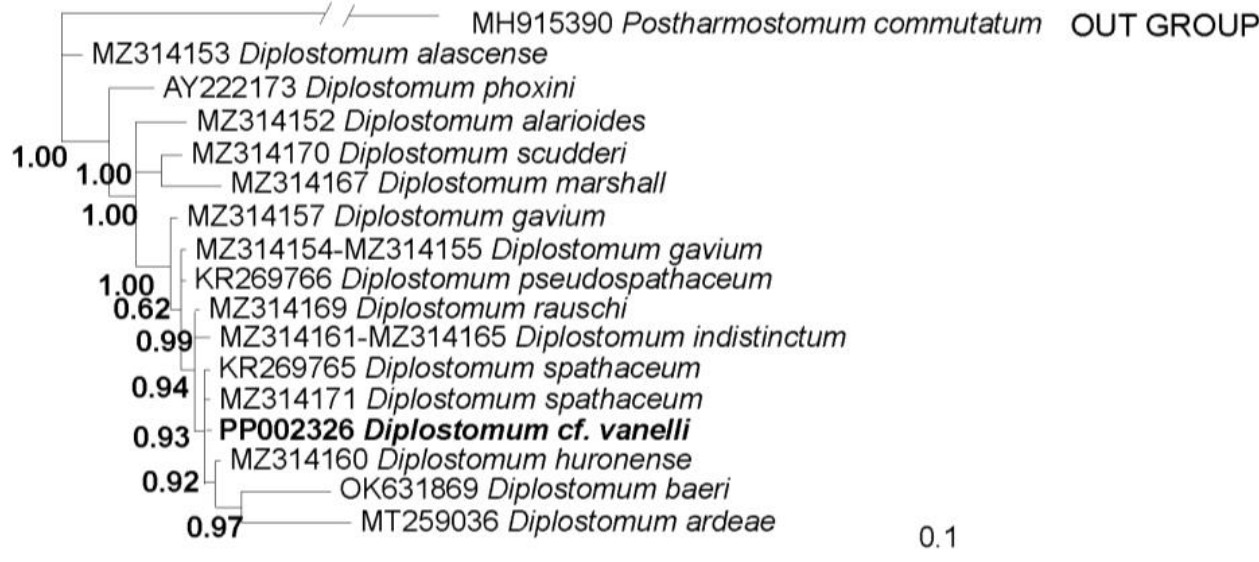

**Figure 2.** Phylogeny of the genera *Diplostomum* based on *28S* sequences using the Bayesian algorithm. A posterior probability of ≥50 was shown in the nodes. The scale bar indicates the number of substitutions per site. Samples from this study in bold.

**Table 5.** Genetic distances between *Diplostomum spathaceum*, *D. cf. vanelli*, *D. huronense*, *D. baeri*, and *D. ardea* based on *28S* sequences. The interspecific distances 0.1–3.6% were obtained in the present study. Standard error in bold.

| | Species | Distances between Species | | | | | Distances within Species | |
|---|---|---|---|---|---|---|---|---|
| | | 1 | 2 | 3 | 4 | 5 | | |
| 1 | *Diplostomum spathaceum* | | **0.0000** | **0.0008** | **0.0044** | **0.0039** | 0.0000 | **0.0000** |
| 2 | *Diplostomum cf. vanelli* | 0.0000 | | **0.0008** | **0.0044** | **0.0039** | - | - |
| 3 | *Diplostomum huronense* | 0.0009 | 0.0009 | | **0.0042** | **0.0038** | - | - |
| 4 | *Diplostomum baeri* | 0.0188 | 0.0188 | 0.0179 | | **0.0049** | - | - |
| 5 | *Diplostomum ardea* | 0.0224 | 0.0224 | 0.0215 | 0.0323 | | - | - |

**Table 6.** Genetic distances between *Diplostomum cf. vanelli*, *D. indistinctum* (Branch 4), and *D. spathaceum*, based on COX1 sequences. The interspecific distances 3.2–15.9% were obtained in the present study. Standard error in bold.

| | Species | Distances between Species | | | Distances within Species | |
|---|---|---|---|---|---|---|
| | | 1 | 2 | 3 | | |
| 1 | *Diplostomum cf. vanelli* | | **0.0126** | **0.0132** | - | - |
| 2 | *Diplostomum spathaceum* | 0.0963 | | **0.0126** | 0.0094 | **0.0021** |
| 3 | *Diplostomum indistinctum* | 0.0849 | 0.0910 | | 0.0082 | **0.0035** |

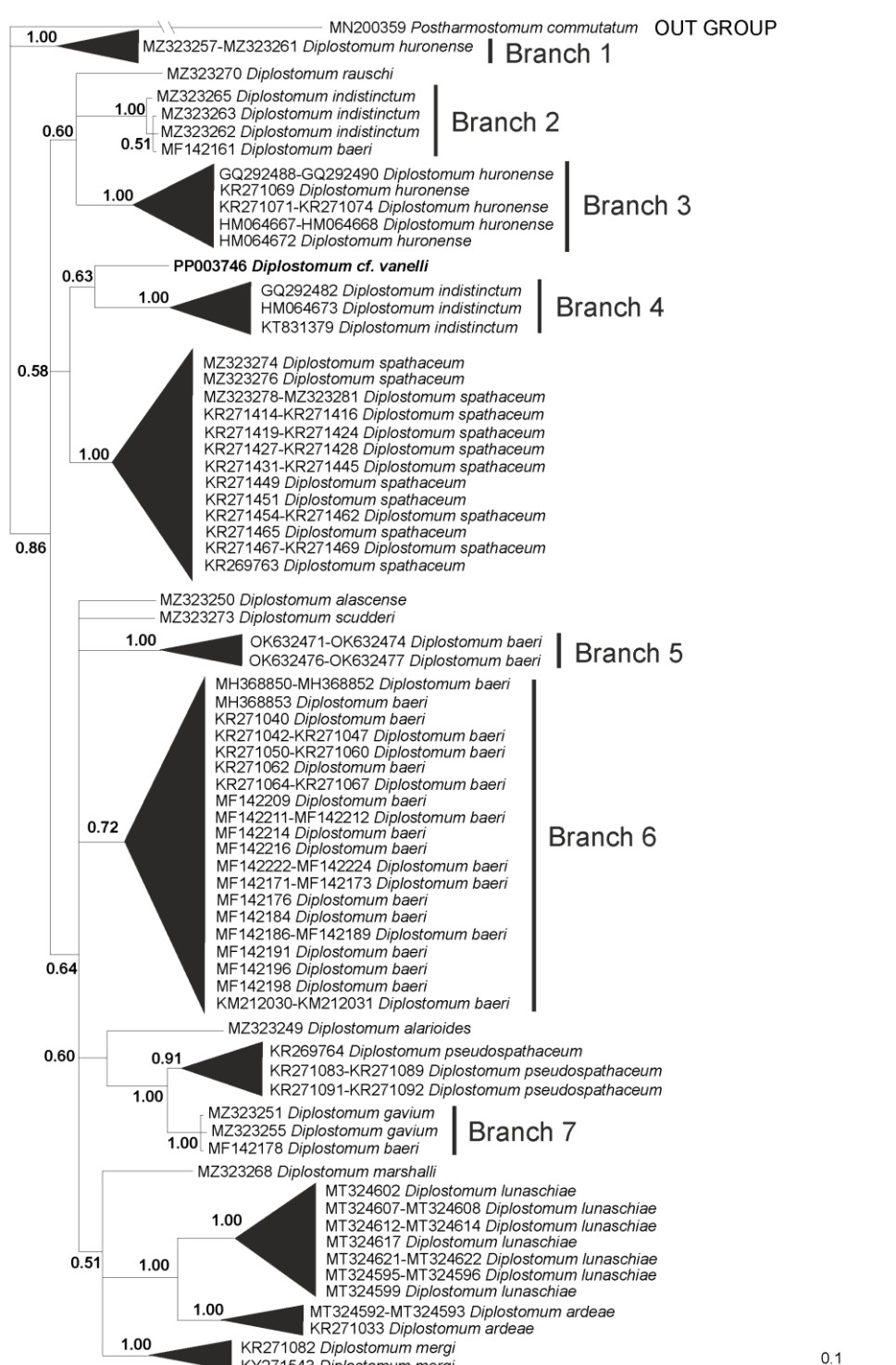

**Figure 3.** Phylogeny of the genera *Diplostomum* based on COX1 sequences using the Bayesian algorithm. A posterior probability of ≥50 was shown in the nodes. The scale bar indicates the number of substitutions per site. Samples from this study in bold.

## 4. Discussion

Adult specimens of *Diplostomum* sp. from the experimental infections in this study are most morphologically similar to *D. kronschnepi*, *D. helveticum*, *D. mergi*, *D. spathaceum*, and *D. vanelli* in the structure of the prosoma and opisthosoma. However, specimens of the first four species differ from the worms in our material in the extent of the vitelline fields in the prosoma: in these four species the vitelline fields reach the level of the anterior end

of the holdfast organ or ventral sucker, or extend slightly in front of the ventral sucker, as compared to the vitelline fields reaching the level of the posterior end of the pharynx of the specimens in our study. Moreover, *D. mergi* has a prosoma that is significantly longer than the opisthosoma, in contrast to the trematode we obtained, in which the prosoma and opisthosoma are equal or similar in length. In addition, the testes of *D. spathaceum* occupy the second half of the opisthosoma, whereas the testes of the fluke from the present study are located in the middle part of the opisthosoma.

The trematodes we obtained experimentally are most similar in morphometry to those of *D. vanelli*, which were first discovered in the lapwing *Vanellus vanellus* (Linnaeus, 1758) in Japan [32]. Our trematodes and those of *D. vanelli* have the following similar morphological characteristics: the ratio of the oral and ventral suckers; the ratio of the sizes of the prosoma and opisthosoma. In contrast, in the description of adult worms of *D. vanelli* given in the works of Yamaguti (1935) and Dubois (1938) (citing [32]), the vitelline fields in the prosoma reach the ventral sucker or, less often, the middle of the prosoma. At the same time, in the figure from Yamaguti (1935) (citing [32]), the vitelline fields reach the lower edge of the pharynx. The same length and localization of vitelline fields were obtained for the worm we studied. In addition, there are some metric differences between the individuals presented in the publications of Yamaguti (1935) (citing [32]) and Dubois (1938) (citing [32]) and in our material, which we attribute to the juvenile stage of the worms (as evidenced by their non-ovigerous state) in the current study (Table 4).

In the phylogenetic reconstruction based on the *28S* nuclear marker (Figure 2), the genetic distances between our sample and *D. huronense*, *D. baeri*, and *D. ardeae* varied from 0.1 to 3.2%, which corresponds to the interspecific distances in the obtained reconstruction (0.1–3.6%). At the same time, the nucleotide sequence of trematode from this study was identical to samples KR269765 and MZ314171, designated as *D. spathaceum*. However, data on morphology are presented only for *D. baeri* [23]. Faltynkova and co-authors [23] have noted that due to weak morphological differences between *Diplostomum* species, erroneous species identification is possible. For correct species determination, an integrated approach is required, namely, the combination of both nucleotide sequences and the morphology of the worm from which these sequences were obtained. Thus, the species identity of the samples designated as *D. huronense*, *D. spathaceum*, and *D. ardeae* may be mistaken. Moreover, it is possible that trematode and *D. spathaceum* (KR269765, MZ314171), which are not different in nuclear marker, are cryptic species. This assumption is confirmed by data on mitochondrial marker. In phylogenetic reconstruction based on COX1 (Figure 3), our sample and *D. spathaceum* clustered into separate branches, and the genetic distance between them was 9.6%, which is in the range of interspecific differences (3.2–15.9%), while differences within the *D. spathaceum* cluster do not exceed 1.9%. Since both nuclear and mitochondrial marker data were obtained from the same samples of these two species (Tables 2 and 3), we can confidently state that they are separate species. The worms in our material also significantly differ from other representatives of the genus-based mitochondrial marker (Figure 3). Thus, based on the combination of morphological characteristics and the similarity of a number of metric indicators (oral sucker, pharynx, ventral sucker, length of testes (Table 4)), and despite minor differences in metric indicators, we suggest that the trematode obtained as a result of the experimental study may belong to species *D. cf. vanelli*, the validity of which assertion is confirmed by a set of data on nuclear and mitochondrial markers. However, to finally resolve the issue, it is necessary to obtain genetic data for this species from the type localization, since previously, for morphologically similar trematodes from Russia and Japan circulating with the participation of birds and designated as one species, differences were identified at the molecular level [33]. Thus, until molecular data for samples *D. vanelli* are obtained from the type localization, the worm we study will be designated as *D. cf. vanelli*.

In addition to the species affilation of *D. cf. vanelli*, we identified paraphyly for three species of *Diplostomum* (*D. huronense*, *D. indistinctum*, and *D. baeri*) when analyzing the phylogenetic reconstruction for the mitochondrial marker (Figure 3). It was established

that Branches 1, 2, and 5 included adult worms of *D. huronense*, *D. indistinctum*, and *D. baeri*, respectively, while Branches 3, 6, and 7 combined the cercarial and metacercarial stages of the same species, *D. huronense*, *D. indistinctum*, and *D. baeri*. Genetic distances between samples designated as *D. huronense* from Branches 1 and 3 were 10.8%, *D. indistinctum* from Branches 2 and 4 differed by 12.1%, and differences between specimens from Branches 2, 5, 6, and 7 designated as *D. baeri* ranged from 10.8 to 12.6%. The listed distances correspond to the interspecific level. Separating the specimens of the same species into different groups was previously mentioned in the work of [21]. In our opinion, it is most likely due to incorrect species identification of trematodes, primarily at the cercariae and metacercarial stages. Taking into account the above, we reiterate the conclusion of Faltynkova et al. 2022 [23] about the need to use a set of morphological and molecular data obtained from the same individuals. All this indicates the need for a significant revision of the data on *Diplostomum* deposited in the NCBI database.

**Author Contributions:** Conceptualization, A.V.I. and V.V.B.; methodology, A.V.I.; software, A.V.I.; validation, V.V.B. and M.Y.S.; investigation, A.V.I.; resources, A.V.I.; data curation, V.V.B.; writing—original draft preparation, A.V.I.; writing—review and editing, V.V.B. and M.Y.S.; visualization, A.V.I.; project administration, V.V.B. and M.Y.S.; funding acquisition, M.Y.S. All authors have read and agreed to the published version of the manuscript.

**Funding:** The research was carried out within the framework of the state assignment from the Ministry of Science and Higher Education of the Russian Federation (theme No. 121031000154-4) and government assignment No. 123022200035-0, "The structure of natural foci of parasitic diseases in the south of the Russian Far East".

**Institutional Review Board Statement:** The authors assert that all procedures contributing to this work comply with the ethical standards of the relevant national and institutional guides on the care and use of laboratory animals including birds and mammals. Euthanasia of all the animals was carried out in accordance with the decision of the Committee on the Ethics of Animal Experiments, Federal Scientific Center of the East Asia Terrestrial Biodiversity (FSCEATB), Far Eastern Branch, Russian Academy of Sciences (FEB RAS) (Permit No. 1 of 25 April 2022).

**Data Availability Statement:** All newly generated sequences were deposited in the GenBank database under the following accession numbers: PP002326 and PP003746.

**Conflicts of Interest:** The authors declare no conflicts of interest.

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
