# Peer review of "Diplostomum cf. vanelli Yamaguti, 1935 (Trematoda: Diplostomidae Poirier, 1886): Morpho-Molecular Data and Life Cycle"

_diversity, doi:10.3390/d16050286_

Round 1

Reviewer 1 Report

Comments and Suggestions for Authors

Reviewer 2 Report

Comments and Suggestions for Authors

 Dear All,

 I really enjoyed reading a study based on an experimental life cycle of a diplostomid digenean, where metacercariae and adults were experimentally reared from cercariae obtained from naturally infected Radix snails. However, as no mature adults were obtained, and due to morphological discrepancies with other species of Diplostomum, as well as similarities in sequences, particularly with those of D. spathaceum, especially when Radix auricularia serves as the host of D. spathaceum in Europe (see Soldanova et al., 2010), I suggest refraining from assigning the specimens obtained herein to D. vanelli. Perhaps they could be treated throughout the manuscript as cf., cfr., confer, or conformis, to avoid future taxonomic problems.

Additionally, I am uncertain if the life cycle of D. vanelli is known. It would be beneficial if the authors explicitly mentioned this. Since there is no comparison provided in the manuscript, it is unclear whether the cercariae of D. vanelli are known. Therefore, it would be advantageous if the cercariae from this study aere compared with those of D. spathaceum and other cryptic species related to these specimens. This comparison would contribute to an integrative approach to species differentiation and aid in understanding the taxonomic position of these specimens. The same applies to the metacercariae.

Abstract:

 Line 12: A space is missing between "a" and "reservoir."

Line 16-17: "Moreover, molecular data for nuclear and mitochondrial markers were also collected." It seems to me that "collected" is not the appropriate verb for obtaining sequences. Although English is not my native language, it appears to me that the manuscript should be reviewed by a native speaker, as there are other problematic parts, such as Line 17-18.

Introduction:

 Line 30: Should "et al." be deleted?

Materials and Methods:

 Line 60-63/Line 66-67: It seems to me that this information should be included in the Results section, while the Materials and Methods section should only describe how they were infected and how often they were reviewed post-infection.

Line 81: "Genomic DNA was extracted from one sample using the HotSHOT technique [14]." Could you clarify what type of worm/stage this corresponds to that was analyzed?

Line 105: It would be beneficial to include a table of the p-distances between and within species in the Results section.

Line 154: "Seminal receptacle located to the left of the ovary." This is not illustrated in the figure.

Line 166: "Anterior end of body armed by 3-4 rows of spines surrounding the oral organ." I cannot see them in the figure. Are they drawn?

Line 191: I would like to request a molecular section in the Results, including a table with distances between sequences and the main results of the Bayesian analysis.

Discussion:

 Line 228: In my opinion, experimental specimens of D. vanelli should be compared more specifically (e.g., measurements) with those of D. spathaceum, given the molecular coincidence. The issue is that, since the specimens do not have eggs, their final size is not known, nor is it known if they completed their development, considering that the experimental host is different from the previously recorded natural host. Furthermore, given the known effect of the host on the morphology of the parasite, I have doubts regarding assigning a specific name to these specimens.

Beside Line 248-249: "And once again, it is worth emphasizing that a morphological description of mature individuals is necessary for these samples from Japan." For this reason, I would not assign a name to these individuals from Japan.

References:

Line 301: Diplostomum should be italicized.

Author Response

Dear colleague,
Thank you for reading our manuscript and making useful comments. We agree with all correction indicated by the reviewer. Below you will find the response to your comments, and we have made changes to the manuscript and marked them using highlighting the text green. 

Abstract:
Line 12: A space is missing between "a" and "reservoir." 
Corrected
Line 16-17: "Moreover, molecular data for nuclear and mitochondrial markers were also collected." It seems to me that "collected" is not the appropriate verb for obtaining sequences. Although English is not my native language, it appears to me that the manuscript should be reviewed by a native speaker, as there are other problematic parts, such as Line 17-18.
All text was checked by Haneef Ahmed Amissah. Mr. Amissah was born in Ghana, which means that he is a native speaker. Moreover, Mr. Amissah is currently a MSc. Candidate in Far Eastern Federal University (Molecular and Cell Biology Program) and in his country, he worked with parasitic infections. Thus, he is the ideal specialist to check the manuscript text in our field.
We have replaced "collected" with "obtained".

Introduction:
Line 30: Should "et al." be deleted?
Corrected

Materials and Methods:
Line 60-63/Line 66-67: It seems to me that this information should be included in the Results section, while the Materials and Methods section should only describe how they were infected and how often they were reviewed post-infection.
Corrected
Line 81: "Genomic DNA was extracted from one sample using the HotSHOT technique [14]." Could you clarify what type of worm/stage this corresponds to that was analyzed?
Corrected
Line 105: It would be beneficial to include a table of the p-distances between and within species in the Results section.
Corrected
Line 154: "Seminal receptacle located to the left of the ovary." This is not illustrated in the figure.
Corrected
Line 166: "Anterior end of body armed by 3-4 rows of spines surrounding the oral organ." I cannot see them in the figure. Are they drawn?
Yes, these spines are indicated in the figure, but due to their size, unfortunately, they are difficult to remove.
Line 191: I would like to request a molecular section in the Results, including a table with distances between sequences and the main results of the Bayesian analysis.
Corrected

Discussion:
Line 228: In my opinion, experimental specimens of D. vanelli should be compared more specifically (e.g., measurements) with those of D. spathaceum, given the molecular coincidence. The issue is that, since the specimens do not have eggs, their final size is not known, nor is it known if they completed their development, considering that the experimental host is different from the previously recorded natural host. Furthermore, given the known effect of the host on the morphology of the parasite, I have doubts regarding assigning a specific name to these specimens.
Considering that the trematodes we obtained had a formed reproductive system and there were no eggs in the uterus, we did not know whether their growth were complete. Therefore, we did not compare the metric data of our samples and other similar diplostome. We agree with your concerns about indicating the species affiliation and add "cf." to the name of the worm we studied.
Beside Line 248-249: "And once again, it is worth emphasizing that a morphological description of mature individuals is necessary for these samples from Japan." For this reason, I would not assign a name to these individuals from Japan.
Corrected

References:
Line 301: Diplostomum should be italicized.
Corrected

Reviewer 3 Report

Comments and Suggestions for Authors

This is a clearly written manuscript on identification and speciation of Diplostomum species. The authors are to be commended on their comparative approach of using both morphology criteria and molecular criteria in the same study. There are a few editorial comments in the edited pdf for the authors to consider. The authors should have a short description of their euthanasia and necropsy technique for the tadpoles, fish and ducklings.  There should also be a description of how the ducklings were fed the metacercaria.

Comments on the Quality of English Language

Comments added to pdf of manuscript

Author Response

Dear colleague,
Thank you for reading our manuscript and making useful comments. We agree with all correction indicated by the reviewer. We have made changes to the manuscript and marked them using highlighting the text green.

Round 2

Reviewer 1 Report

Comments and Suggestions for Authors

The corrections and clarifications on this manuscript are acceptable and improve the presentation and clarity.

Seven very minor edits for tense etc are noted in the attached.

Please check journal format for COX1 vs cox1

Very nice paper.

Comments on the Quality of English Language

Author Response

Good afternoon, thank you for your edits, we have made all the necessary adjustments.
